# Host-parasite tissue adhesion by a secreted type of β-1,4-glucanase in the parasitic plant *Phtheirospermum japonicum*

Ken-ichi Kurotani[1,10], Takanori Wakatake[2,3,8,10], Yasunori Ichihashi[3,9], Koji Okayasu[4], Yu Sawai[4], Satoshi Ogawa[3], Songkui Cui [5], Takamasa Suzuki [6], Ken Shirasu[2,3 ✉] & Michitaka Notaguchi [1,4,7 ✉]

Tissue adhesion between plant species occurs both naturally and artificially. Parasitic plants establish intimate relationship with host plants by adhering tissues at roots or stems. Plant grafting, on the other hand, is a widely used technique in agriculture to adhere tissues of two stems. Here we found that the model Orobanchaceae parasitic plant *Phtheirospermum japonicum* can be grafted on to interfamily species. To understand molecular basis of tissue adhesion between distant plant species, we conducted comparative transcriptome analyses on both infection and grafting by *P. japonicum* on *Arabidopsis*. Despite different organs, we identified the shared gene expression profile, where cell proliferation- and cell wall modification-related genes are up-regulated. Among genes commonly induced in tissue adhesion between distant species, we showed a gene encoding a secreted type of β-1,4-glucanase plays an important role for plant parasitism. Our data provide insights into the molecular commonality between parasitism and grafting in plants.

[1] Bioscience and Biotechnology Center, Nagoya University, Furo-cho, Chikusa-ku, Nagoya 464-8601, Japan. [2] Graduate School of Science, The University of Tokyo, Bunkyo-ku, Tokyo 113-8657, Japan. [3] Center for Sustainable Resource Science, RIKEN, Tsurumi, Yokohama, Kanagawa 230-0045, Japan. [4] Graduate School of Bioagricultural Sciences, Nagoya University, Furo-cho, Chikusa-ku, Nagoya 464-8601, Japan. [5] Institute for Research Initiatives, Division for Research Strategy, Nara Institute of Science and Technology, Ikoma, Nara 630-0192, Japan. [6] College of Bioscience and Biotechnology, Chubu University, Matsumoto-cho, Kasugai 487-8501, Japan. [7] Institute of Transformative Bio-Molecules, Nagoya University, Furo-cho, Chikusa-ku, Nagoya 464-8601, Japan. [8] Present address: Biocenter, Institute for Molecular Plant Physiology and Biophysics, Julius-von-Sachs-Institute, University of Würzburg, 97082 Würzburg, Germany. [9] Present address: RIKEN BioResource Research Center, Tsukuba, Ibaraki 305-0074, Japan. [10] These authors contributed equally: Ken-ichi Kurotani, Takanori Wakatake. ✉email: ken.shirasu@riken.jp; notaguchi.michitaka@b.mbox.nagoya-u.ac.jp

Exceptionally in the autotrophic plant lineage, parasitic plants have evolved the capability to absorb water and nutrients from other plants. This ability relies on a specialized organ called a haustorium, which forms a physical and physiological connection between the parasite and host tissues[1]. Plant parasitism has independently evolved in angiosperm lineages at least 12 times and ~1% of angiosperms are estimated to be parasitic[2,3]. Among these species, the Orobanchaceae family is the most species-rich and includes the notorious agricultural pests *Striga* spp., *Phelipanche*, and *Orobanche* spp., which threaten world food security[4].

The infection process of parasitic plant to host plant tissues is initiated with physical contact between the parasitic haustorium and the host tissue. After recognizing the host plant by the acceptance of host-derived compounds, the parasitic plants promote the development of haustorium and haustorial hairs derived from epidermal cells. Then, adhesion between parasitic haustorium and host plants is established. In the haustorial hair defective mutants, the number of haustoria formed upon infection of the host roots is reduced, but the internal structure of haustoria remains intact[5]. Electron micrographs of the interaction between *Striga* haustorium and a host show that parasitic invasion is accompanied by host cell wall alterations, but not disruption, such as partial wall dissolution and shredding[6]. Similarly, *Orobanche* spp. penetrate the host root tissues where pectolytic enzyme activity is evident around haustoria[7]. Activities of cell wall-degrading enzymes, such as cellulase and polygalacturonase, are also present in infecting *Phelipanche* tubers[8]. In the case of stem parasites, such as dodder (*Cuscuta pentagona*), epidermal cells differentiate into secretory trichomes that excrete a cementing substance predominantly composed of de-esterified pectins, and the cell walls are modified by a cell wall-loosening complex[9]. The parasitic haustorium thus is able to adhere to the host tissues either in roots or in stems.

All the parasitic plants known to date are able to establish vasculature connection to host, which can be considered as "natural grafting". Especially, one of the interesting characteristics of parasitic plants is their ability to adhere to the apoplastic cell wall matrix of phylogenetically distant plant species of diverse cell wall composition. This adhesion ability is also crucial for "artificial grafting", in which cut stem tissues are assembled to unite, and often causes incompatibility among interfamily species[10]. In the case of compatible graft combinations, the grafted parts are connected through tissue adhesion. Compressed cell walls in the region of the graft junction have been observed during grafting[11,12], which indicates that the cell walls between opposing cells at the graft interface adhere followed by vascular reconstruction and tissue union between the grafted organs. The mechanism of how parasitic plants are able to overcome incompatibility in tissue adhesion with a diverse range of host plant species remains unclear.

To understand molecular events during parasite infection, transcriptome analyses have been conducted on several parasitic plants, including species of Orobanchaceae, as well as dodder[13–18]. In particular, Yang et al. identified putative parasitism genes that are upregulated during haustorial development following host attachment in three Orobanchaceae parasitic species[13]. Among them, genes that encode proteases, cell wall-modification enzymes, and extracellular secretion proteins are highly upregulated. Similarly, transcriptome analysis of dodder revealed increased expression of genes encoding cell wall-modifying enzymes, such as pectin lyase, pectin methyl esterase, cellulase, and expansins, in the infective stages[14]. A transcriptome analysis of *Thesium chinense*, a parasitic Santalaceae plant, also identified highly upregulated genes that encode proteins associated with cell wall organization as a peripheral module in the gene co-expression network during

developmental reprogramming of haustorium[19]. In addition, upregulation of genes that encode cell wall-modifying enzymes was detected in the transcriptome of host–parasite interface in the model *Triphysaria versicolor*, using laser microdissection[20]. These aforementioned results suggest that parasitic plants facilitate cell–cell adhesion at the interface between the haustorium and host through activation of cell wall-modification enzymes.

In this study, we addressed molecular commonality between parasitic infection and artificial grafting by comparing tissue adhesion events between *Phtheirospermum japonicum* and *Arabidopsis*. Although these events occur in different organs, we expected that key common components for tissue adhesion would be found by comparative transcriptomic analyses. In addition, we further compared these datasets with that of interfamily graft of *Nicotiana benthamiana*, which is able to adhere cells with those of plant species from diverse families in grafting[21]. We identified nine genes that were commonly upregulated in *P. japonicum* haustorium, *P. japonicum/Arabidopsis,* and *N. benthamiana/Arabidopsis* grafting sites. Among them, we identified a gene encoding β-1,4-glucanase as an important factor in plant parasitism.

## Results

**Tissue adhesion between *P. japonicum* and *Arabidopsis* in parasitism and grafting.** A facultative parasitic plant, *P. japonicum*, has been studied previously as a model root parasite that can parasitize *Arabidopsis*[16,22,23]. The ability to transport materials from *Arabidopsis* to *P. japonicum* can be visualized using a symplasmic tracer dye, carboxyfluorescein (CF)[23] (Fig. 1a, b). At the parasitization site in the root, a xylem bridge is formed in the haustorium (Fig. 1c), by which the *P. japonicum* tissues invade the host tissues (Fig. 1d). We observed the interface of the *P. japonicum* haustorium and *Arabidopsis* root tissues using transmission electron microscopy (Fig. 1e–i). The cells at the tip of the penetrating haustoria adhered closely to the opposing *Arabidopsis* cells where thin cell walls were observed (Fig. 1e). Serial sections revealed a decrease in cell wall thickness at the interface between *P. japonicum* and *Arabidopsis* tissues (Fig. 1f–i), which indicated that cell wall digestion occurred at the interface.

We observed similar thin cell walls at graft boundary between *Arabidopsis* and *Nicotiana* species, which exhibit a capability to adhere their tissue across interfamily species[21,24] (Fig. 2a, b). Therefore, we hypothesized that the parasitic plant may also have a wide tissue adhesion capability in artificial grafting. To test this hypothesis, we grafted a stem of *P. japonicum* (as the scion) onto the *Arabidopsis* inflorescence stem. Cells proliferated on the cut surface of the scion, similar to haustorial tissues. The *P. japonicum* scion was able to establish a graft union with the *Arabidopsis* stock and remained viable for 1 month after grafting (Fig. 2c). Given that parasitic Orobanchaceae species have a diverse host range among angiosperms[25], we further tested graft combinations using nine species from seven orders of angiosperms. The grafting capability of *P. japonicum* as the scion using these interfamily species as the stock was clearly observed, except for two Fabaceae species (Fig. 2d–f and Supplementary Data 1). Reciprocally, *P. japonicum* was able to be used as the stock plant for certain plant species (Fig. 2g and Supplementary Data 1). In contrast, when *Lindenbergia philippensis*, which has no parasitic ability among Orobanchaceae[4], was grafted onto *Arabidopsis*, viability of the *L. philippensis* scion was extremely limited compared with the *P. japonicum* scion; nine graft trials were never successful, although all nine homografts of *L. philippensis* were successful (Supplementary Data 1). When we observed cross-sections of the graft junction of *P. japonicum/Arabidopsis* (scion/stock), xylem continuity, and apoplastic dye transport were observed (Fig. 2h, i). Importantly, establishment of the

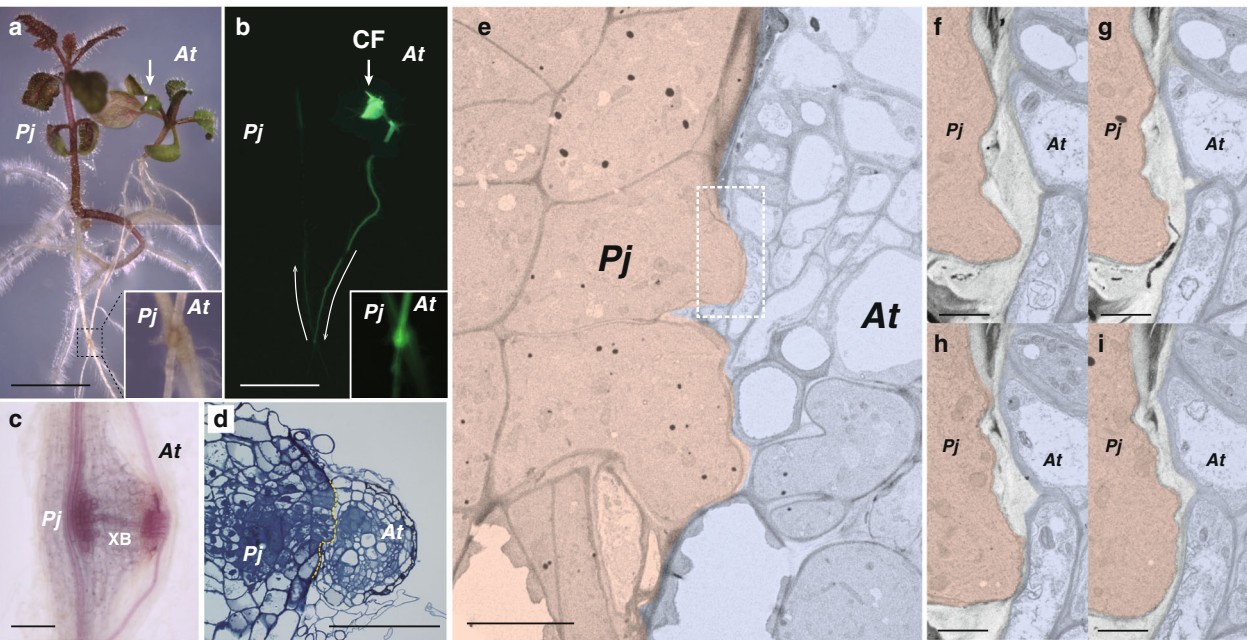

**Fig. 1 Parasitism of *P. japonicum*. a, b** Parasitism between the roots of *P. japonicum* and *Arabidopsis*. The *P. japonicum* root parasitized the *Arabidopsis* root (insets). Transport of a symplasmic tracer dye, carboxyfluorescein (CF, colored in green), showed establishment of a symplasmic connection between the plants; light micrograph (**a**) and fluorescence image (**b**). Arrows indicate the site where CF dye was applied and the direction of transport. **c, d** Site where *P. japonicum* parasitized the *Arabidopsis* root. **c** Phloroglucinol staining showing xylem bridge formation (XB). **d** Cross-section of the parasitization site. The *P. japonicum* tissue invaded the *Arabidopsis* root tissues. Dashed line indicates the interface of parasitism. **e** Transmission electron micrograph of the interface between *P. japonicum* (pink) and *Arabidopsis* (blue). Partial tissue adhesion was observed at the interface. The dashed rectangle indicates the area of (**f–i**). **f–i** Serial sections at the interface between *P. japonicum* and *Arabidopsis* cells. The cell wall was partially digested. *Pj P. japonicum, At Arabidopsis*. Scale bars, 5 mm (**a**, **b**), 100 μm (**c**, **d**), 10 μm (**e**), and 2 μm (**f–i**).

symplast between *P. japonicum* and *Arabidopsis* was confirmed by using the CF dye (Fig. 2j). In summary, these results showed that the root parasite *P. japonicum* is able to achieve tissue adhesion and vasculature connection with members of diverse plant families in both parasitism and grafting.

**Transcriptome analyses of parasitism and grafting**. To investigate molecular events involved in cell–cell adhesion between *P. japonicum* and the host plant, we analyzed the transcriptome for *P. japonicum*–*Arabidopsis* parasitism and *P. japonicum*/*Arabidopsis* grafting. For this purpose, taking into account the periods during which parasitism and grafting between distantly related plants are established, sequential samples of haustorial infection sites at the roots from 1 to 7 days post infection (DPI) and of grafted region on the stems from 1 to 14 days after grafting (DAG) were collected and subjected to RNA-sequencing (RNA-Seq) analysis. The obtained sequence reads were mapped on the genome references of *P. japonicum* or *Arabidopsis* and uniquely mapped reads to the *P. japonicum* reference were further analyzed (see detail in "Methods"). Preliminary analysis of these datasets by principal component analysis (PCA) indicated that the transcriptomes of *P. japonicum*–*Arabidopsis* parasitism of the root and *P. japonicum*/*Arabidopsis* grafting onto the shoot differed substantially. The two transcriptomes showed a relatively similar distribution on PC1, but were widely separated on PC2 (Fig. 3a). Hierarchical clustering also indicated that gene expression patterns during parasitism and grafting were different over all (Fig. 3b). Numerous genes were highly upregulated during parasitism or grafting, including some genes previously known to be associated with wound healing processes, such as

auxin action, procambial activity, and vascular formation[19,26,27] (Fig. 3b, c). However, the expression of many genes was distinct between parasitism and grafting. For example, *PIN1*, which encodes an auxin efflux transporter, *cyclin B1;2*, a cell-cycle regulator, *PLL1*, involved in maintenance of the procambium, *VND7*, a NAC domain transcriptional factor that induces trans-differentiation of various cells into protoxylem vessel elements, and *OPS*, a regulator of phloem differentiation, were all upregulated in both parasitism and grafting, even though the individual daily patterns were not exactly same probably because of different phenomena in different organs. In contrast, *IAA1*, which encodes an auxin-induced regulator, *ANAC071*, a transcriptional factor involved in tissue reunion after wounding, and *LBD16* and *LBD29*, LOB domain proteins involved in lateral root development, were upregulated only in grafting but not in parasitism. Conversely, *WOX4*, which encodes a WUSCHEL-related homeobox protein maintaining cambium activity, and *TMO6*, which encodes a Dof-type transcriptional factor for vascular development, were upregulated only in parasitism but not in grafting. The expression levels of *IAA18*, which encodes an auxin-induced regulator, *PLT3* and *PLT5*, which encode AP2-domain transcription factors involved in root stem cell identity, *CESA4*, which encodes cellulose synthase involved in secondary cell wall biosynthesis, and *ALF4*, involved in lateral roots initiation, showed little change or tended to decrease in both parasitism and grafting.

To identify common molecular events, we generated clusters for parasitism and grafting by multivariate analysis using self-organized map clustering, and compared clusters that included genes upregulated during tissue adhesion in parasitism and grafting (Fig. 4a, b and Supplementary Fig. 1). During *P. japonicum*

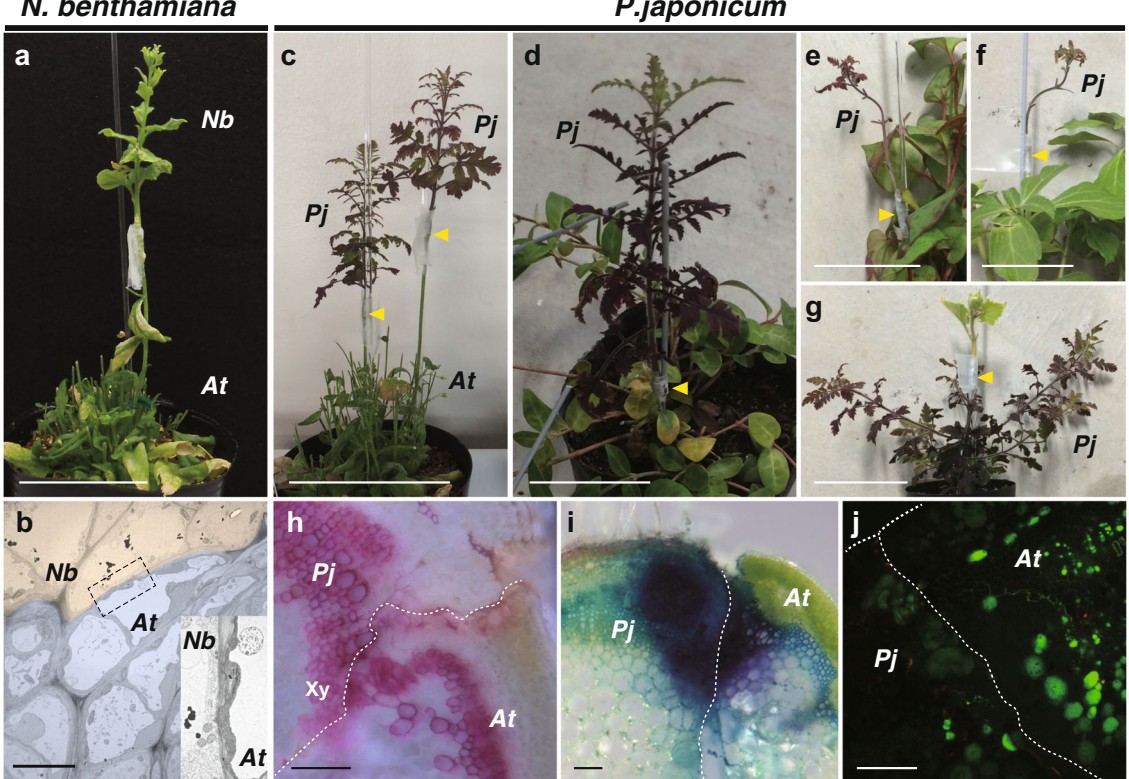

**Fig. 2 Heterospecific grafting of N. benthamiana and P. japonicum. a** Grafting of *N. benthamiana* onto *Arabidopsis* at 28 days after grafting (DAG). **b** Transmission electron micrograph of the graft junction between *N. benthamiana* (pink) and *Arabidopsis* (blue). Inset represents a magnified image of the area of the dashed rectangle. **c**–**g** Grafting of *P. japonicum* with diverse plant species; grafts of *P. japonicum* as the scion onto stems of *Arabidopsis* at 28 DAG (**c**), *Vinca major* at 52 DAG (**d**), *Houttuynia cordata* at 30 DAG (**e**), and *Pachysandra terminalis* at 30 DAG (**f**), and graft of *Cucumis sativus* as the scion onto *P. japonicum* stock at 30 DAG (**g**). Arrowheads indicate grafting points. **h** Cross-section of the graft junction of *P. japonicum*/*Arabidopsis*. Phloroglucinol staining showing xylem formation in the graft region (Xy). **i** Cross-section of the graft junction of *P. japonicum*/*Arabidopsis* stained with toluidine blue applied to the *Arabidopsis* stock at 14 DAG. Detection of dye transport from *Arabidopsis* to *P. japonicum* demonstrated establishment of apoplastic transport. **j** Symplasmic transport establishment was confirmed using carboxyfluorescein (CF). CF was applied in the diacetate form to the leaves of the *Arabidopsis* stock and a cross-section of the graft junction of *P. japonicum*/*Arabidopsis* was observed. The CF fluorescence was detected in *P. japonicum* tissues. *Pj* *P. japonicum*, *At Arabidopsis*. Scale bars, 5 cm (**a**, **c**–**g**), 1 µm (**b**), and 100 µm (**h**–**j**).

parasitization of the host root, tissue adhesion between *P. japonicum* and the host occurred around 1–2 DPI, and then a xylem bridge connecting a *P. japonicum* root vessel and a host vessel was formed at 3 DPI (Fig. 4a). In contrast, histological observation showed that tissue adhesion between the scion and stock during grafting occurred about 3 DAG (Fig. 4b). We focused three clusters with distinct expression patterns (Fig. 4a, b). The first one includes genes upregulated during the tissue adhesion stage starting about 1 DPI or 1 DAG (Node 09 for parasitism, Node 10 for grafting). The second one contains genes upregulated along the time (Node 05 for parasitism and Node 08 for grafting). The third one includes genes peaked around 2 DPI or 3 DAG (Node 08 for parasitism and Node 11 for grafting). Gene ontology (GO) enrichment analysis revealed many of the enriched GO terms were common to parasitism and grafting (Supplementary Fig. 1 and Supplementary Data 4–6). Especially in the focused three clusters, many of the enriched GO terms were associated with cell division, such as DNA replication, cytoplasm, and RNA metabolism, which reflected the occurrence of active cell proliferation in the haustorium and the graft interface. Importantly, these clusters also included enriched GO terms associated with the cell wall common to parasitism and grafting (Fig. 4c and Supplementary Data 7–10).

To identify genes specifically associated with tissue adhesion, we further compared these data with transcriptome data from grafting between *Nicotiana* and *Arabidopsis*, since a previous study has shown that *Nicotiana* is capable of graft adhesion with a diverse range of angiosperms and this transcriptome was highly enriched with GOs associated with cell adhesion[21] (Fig. 5). We identified nine genes commonly upregulated during tissue adhesion period in the three distinct experiments, including genes associated with cell division, such as cyclin D, and cell wall-related genes, such as glycosyl hydrolase (Fig. 5). One of the identified glycosyl hydrolases belongs to the *Glycosyl hydrolase 9B* (*GH9B*) family, which includes genes encoding β-1,4-glucanases in plants[28,29]. Interestingly, among the *GH9B* family, a member of *GH9B3* clade was recently shown to be crucial for cell–cell adhesion in *Nicotiana* interfamily grafting[21].

**PjGH9B3 is essential for P. japonicum parasitism**. As cell walls locating at the interface between *P. japonicum* and *Arabidopsis* were partially digested (Fig. 1e, f), we decided to analyze function of the *GH9B3* in parasitism. We reconstructed a phylogenetic tree for the *GH9B* gene family for *Arabidopsis*, *P. japonicum*, and *S. hermonthica*, as well as *L. philippensis*, a nonparasitic Orobanchaceae species[30] (Fig. 6). In the phylogenetic tree, five and four genes from *P. japonicum* and *S. hermonthica* are found in the *GH9B3* clade, respectively, while only two and one *GH9B3* genes are present in *Arabidopsis* and *L. philippensis*, respectively

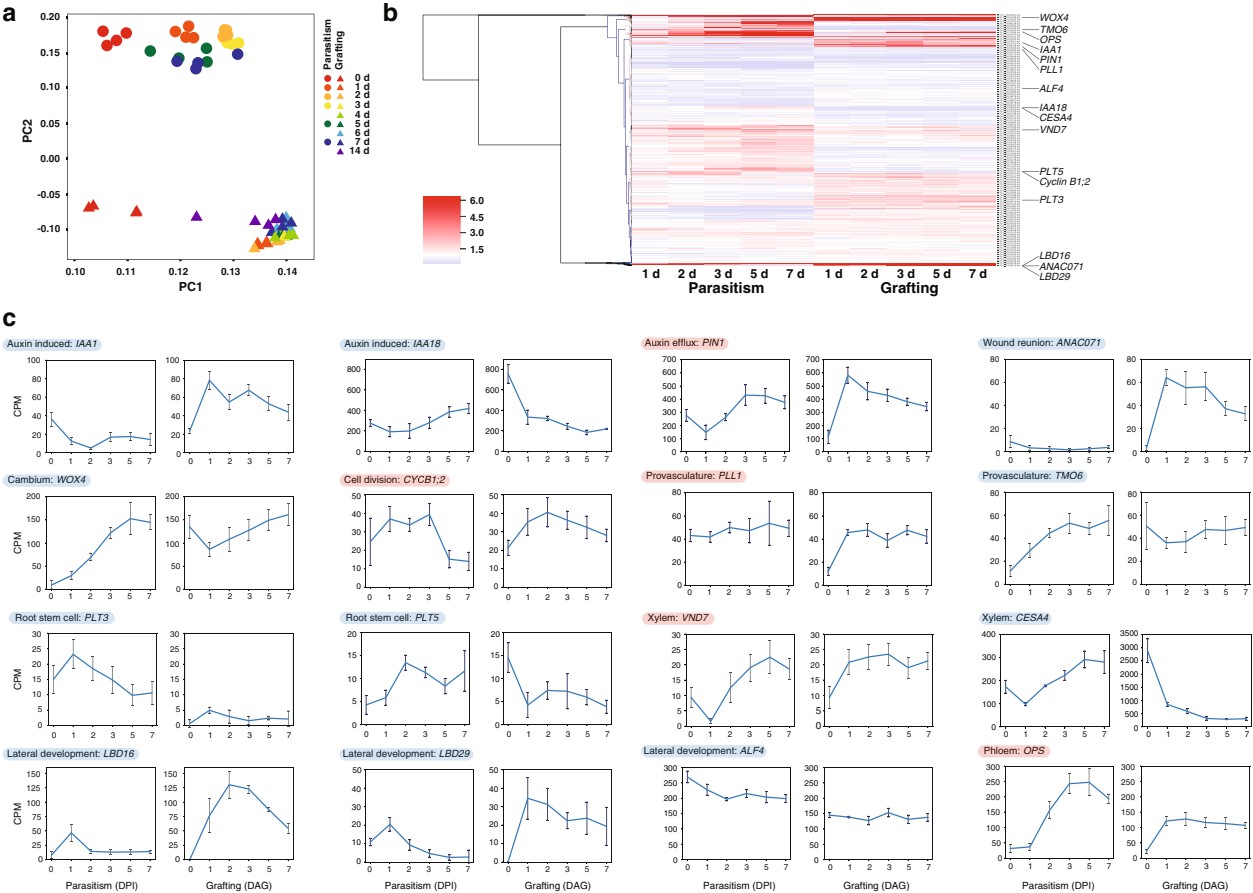

**Fig. 3 Transcriptomic analysis of parasitism and grafting between *P. japonicum* and *Arabidopsis*. a** Transcriptomic analysis was performed using RNA samples of the *P. japonicum* infected site and the *P. japonicum* graft site. For parasitism, total RNA was extracted before and 1, 2, 3, 5, and 7 days post infection (DPI). For grafting, total RNA was extracted before and 1, 2, 3, 4, 5, 6, 7, and 14 days after grafting (DAG) (with four biological replicates for each time point). Principal component analysis was performed from the obtained expression profile. Triangles and circles represent parasitism and grafting, respectively. PC, principal component. **b** Hierarchical clustering using Euclidean distance and Ward's minimum variance method over ratio of RNA-seq data from five time points for *P. japonicum*–*Arabidopsis* parasitism and *P. japonicum*/*Arabidopsis* grafting against intact plants. Genes for which association with parasitism and grafting has been reported in previous studies are marked. Using the cDNA sequence of *Arabidopsis* as queries, tblastx was used to determine the most closely related homologs of *P. japonicum*. **c** Plots of the gene expression levels marked in **b**. Red background represents genes that behaved similarly in parasitism and grafting, and blue background indicates genes that behaved differently.

(Fig. 6a). Moreover, the tendency that the number of genes belonging to the *GH9B3* clade were increased only in the parasitic plant was also found in the phylogenetic trees for *Striga asiatica*, another species of the genus *Striga*, and *Erythranthe guttata* (*Mimulus guttata*), a nonparasitic plant of the *Lamiaceae* (Supplementary Fig. 2a, b). The *LiPhGnB1_1726*, a member of the *L. philippensis GH9B3* clade was upregulated at 1 DAG, but such upregulation was observed even in incompatible graft combinations using other plant species; soybean (*Glycine max*), morning glory (*Ipomoea nil*), and *Arabidopsis*[21]. However, expression did not continue to increase subsequently, as this is often the case for incompatible graft combinations[21] (Fig. 6b). By contrast, *Pjv1_G00028629*, the most similar *P. japonicum* homolog of *NbGH9B3* gene that was associated with grafting, was upregulated at 1 DAG and gradually increased until 7 DAG in grafting, as well as 7 DPI in parasitism (Fig. 6b). Similarly, in *S. hermonthica*, the corresponding homolog *Sh14Contig_25152* was upregulated at 1 DPI with a peak at 3 DPI[18] (Fig. 6b). Some of the other homologous genes of the *GH9B3* clade were also upregulated in parasitism, but not for those of the other *GH9* clades (Supplementary Fig. 3). These data suggest that upregulation of *GH9B3* homologs in parasitic plants at the site of infection and interfamily grafting is conserved in parasitic Orobanchaceae plants.

*Pjv1_G00028629* is upregulated at the early phase of infection and is phylogenetically closest to the *GH9B3* genes of *Nicotiana* and *Arabidopsis*. Alignment analysis of amino acid sequences shows that four genes including *Pjv1_G00028629* conserve a catalytic domain and *O*-glycosylation sites (Supplementary Fig. 4). Therefore, we investigated the function of *Pjv1_G00028629*, called *PjGH9B3*. First, temporal and spacial expression patterns of *PjGH9B3* were examined using a promoter-Venus construct. *PjGH9B3* promoter activity was detected at the cell periphery of the haustorium attaching to the host at 1–2 DPI (Fig. 7a, b), while the signal was later shifted to the inside of haustorium at 3–4 DPI (Fig. 7c, d). These expression patterns indicated that *PjGH9B3* functions at the interface with the host root tissue at the early adhesion stage and xylem formation at the later stage. *PjGH9B3*-knockdown experiments by RNA interference (RNAi) system revealed that *PjGH9B3*-knockdown did not affect haustorium emergence but resulted in significantly fewer successful xylem connections with host, compared with the control (Fig. 7e–i). Reduction of *PjGH9B3* expression was confirmed in hairy roots using quantitative reverse-transcription PCR (RT-qPCR) (Fig. 7f). These data indicate that *PjGH9B3* positively regulates infection processes in *P. japonicum*.

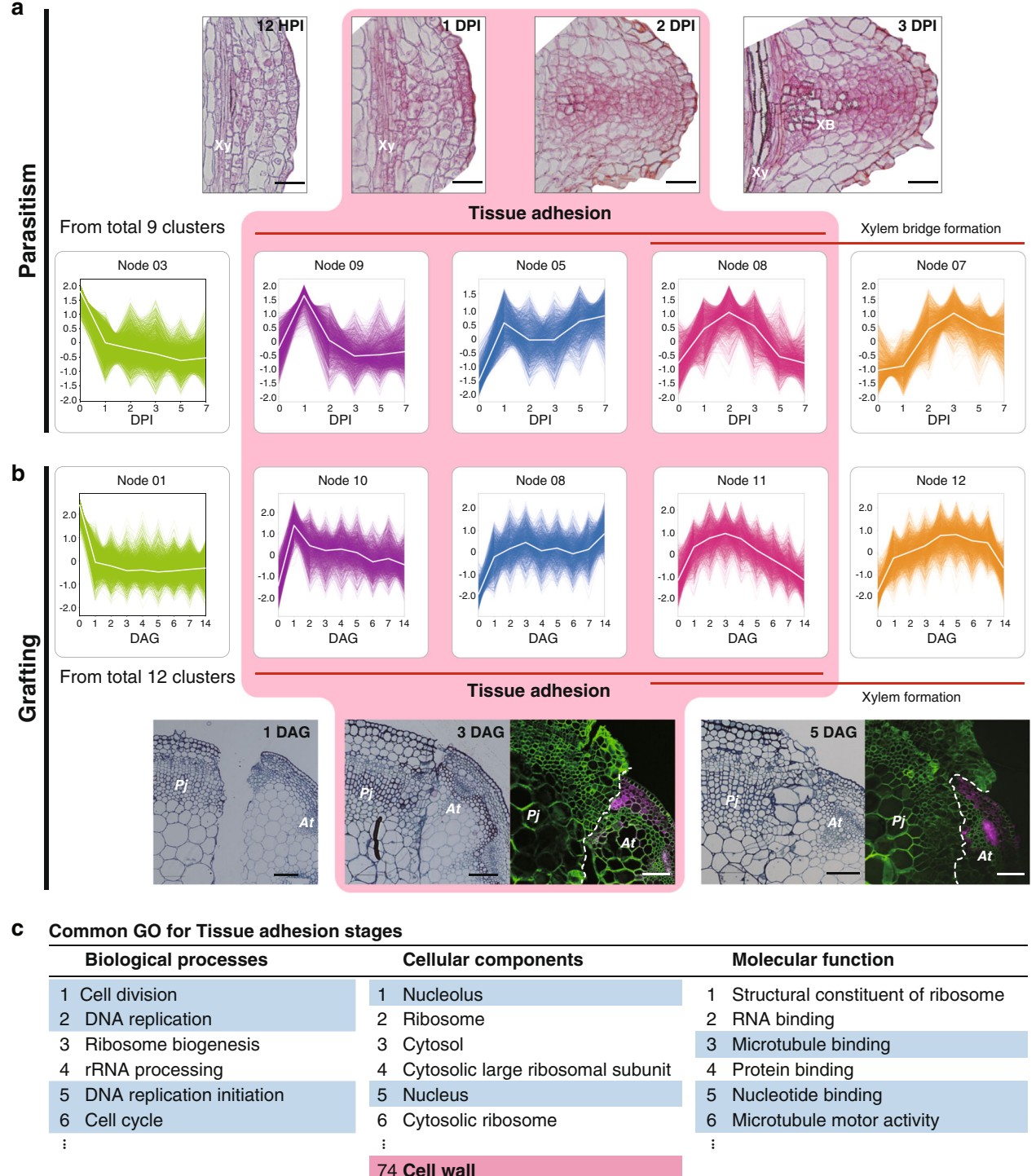

**Fig. 4 Comparison of self-organizing map (SOM) clusters associated with the tissue adhesion stage during parasitism and grafting. a, b** Tissue sections of the parasitized site of *P. japonicum* 12 h post infection (HPI), 1, 2, and 3 DPI (**a**) and the graft junction between *P. japonicum* (*Pj*) and *Arabidopsis* (*At*) 1, 3, and 5 DAG (**b**). Fluorescence images of the graft junction are also shown where *P. japonicum* was grafted onto *Arabidopsis* harboring *RPS5a::LTI6b-tdTomato* (**b**). Green indicates the cell wall, magenta indicates tdTomato fluorescence. Xy xylem, XB xylem bridge. SOM clusters with similar patterns in parasitism (**a**) and grafting (**b**) are shown. **c** Enriched gene ontology (GO) terms common to parasitism and grafting are listed for the clusters. Nodes 5, 8, and 9 were merged from the SOM cluster of the *P. japonicum* parasitism transcriptome, and nodes 8, 10, and 11 were merged from the SOM cluster of the *P. japonicum* grafting transcriptome. A portion of the GO terms categorized as "biological processes", "cellular components", and "molecular functions" are shown. GO terms potentially associated with "cell division" and "cell wall modification" are marked in blue and red, respectively. Scale bars, 50 μm (**a**), and 100 μm (**b**).

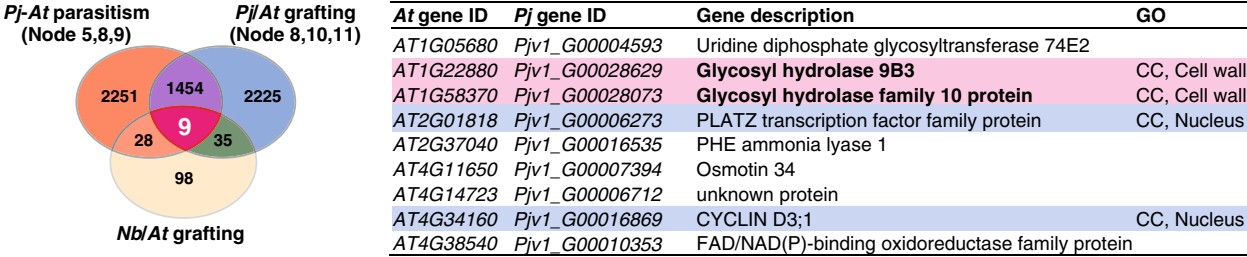

| *At* gene ID | *Pj* gene ID | Gene description | GO |
|---|---|---|---|
| AT1G05680 | Pjv1_G00004593 | Uridine diphosphate glycosyltransferase 74E2 | |
| AT1G22880 | Pjv1_G00028629 | **Glycosyl hydrolase 9B3** | CC, Cell wall |
| AT1G58370 | Pjv1_G00028073 | **Glycosyl hydrolase family 10 protein** | CC, Cell wall |
| AT2G01818 | Pjv1_G00006273 | PLATZ transcription factor family protein | CC, Nucleus |
| AT2G37040 | Pjv1_G00016535 | PHE ammonia lyase 1 | |
| AT4G11650 | Pjv1_G00007394 | Osmotin 34 | |
| AT4G14723 | Pjv1_G00006712 | unknown protein | |
| AT4G34160 | Pjv1_G00016869 | CYCLIN D3;1 | CC, Nucleus |
| AT4G38540 | Pjv1_G00010353 | FAD/NAD(P)-binding oxidoreductase family protein | |

**Fig. 5 Extraction of factors important for tissue adhesion in parasitism and grafting.** Venn diagrams of the three gene populations were plotted, together with 170 *Arabidopsis* genes annotated by 189 *N. benthamiana* genes for which expression was elevated in the early stage of interfamily grafting of *N. benthamiana* and *Arabidopsis*[21]. Nine genes common to the three gene datasets are listed. Genes potentially associated with "cell division" and "cell wall modification" are marked in blue and red, respectively. CC represents the GO subcategories, cellular components.

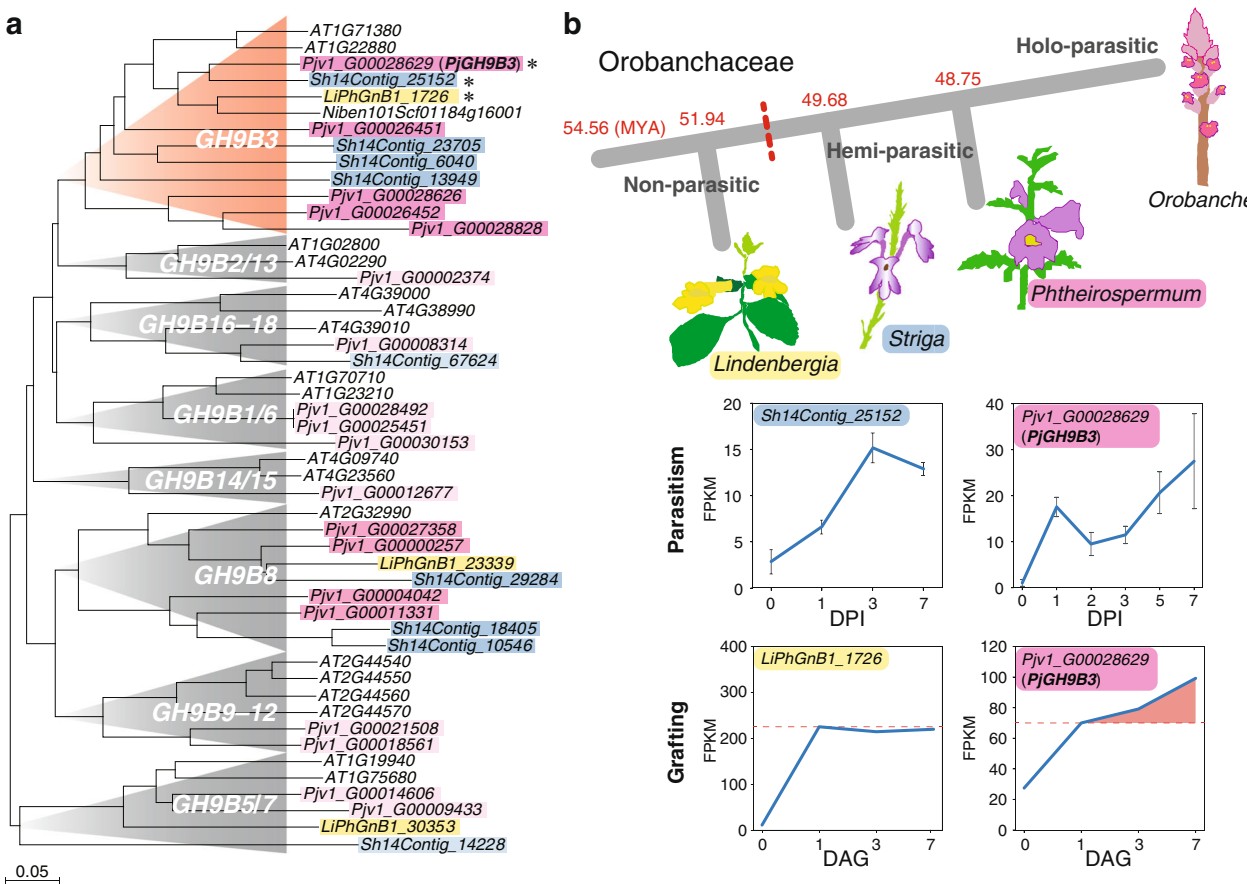

**Fig. 6 *Glycosyl hydrolase 9B* gene family in Orobanchaceae. a** Phylogeny of *Glycosyl hydrolase 9B3* genes of *L. philippensis* (yellow), *P. japonicum* (red), *S. hermonthica* (blue), *Arabidopsis*, and *N. benthamiana* reconstructed using deduced amino acid sequences. If the number of *P. japonicum* or *S. hermonthica* genes in a clade is less than twice the number of *Arabidopsis* genes in the same clade, the gene is highlighted in light pink or light blue. **b** Phylogeny of Orobanchaceae including non-, hemi-, and holo-parasitic species. Estimated branching time points (million years ago, MYA) are indicated[30]. Expression patterns of genes encoding the amino acid sequence closest to *NbGH9B3* and *AtGH9B3* in grafting for *L. philippensis* and *P. japonicum*, in which *Arabidopsis* was used as the stock plants, or parasitism for *S. hermonthica*[18] and *P. japonicum*, in which rice and *Arabidopsis* were the host plants, respectively.

## Discussion

Parasitic invasion by haustorium is established through the disruption of host cell wall barriers following attachment of haustorium to the host tissues by haustorial hairs[1,5,6]. Similar modification of the cell wall is also observed in the graft, which is an artificial tissue connection. In general, grafting is successful between closely related plants, such as members of the same species, genus, and family, but not between genetically distant plants, such as members of different families, with a few exceptions, such as *Nicotiana*, which we previously reported[11,12,21,31,32]. Since the adhesion of plant tissues of different families also occurred in parasitism, we expected

that there might be a commonality between the capability of these parasitic plants to parasitize diverse hosts and their capability to implement heterografting. In support of our assumption, interfamily grafting of *P. japonicum* showed its compatibility with multiple autotrophic plants (Fig. 2c–g and Supplementary Data 1). Because previous studies in *N. benthamiana* displayed that a higher success rate of grafting could be achieved when it was used as a scion rather than a stock[21], we grafted the *P. japonicum* plant mainly as a scion in this study. Remarkably, grafting capability of *P. japonicum* was also observed when it was grafted as a stock onto several plant species, indicating the fundamentally high grafting

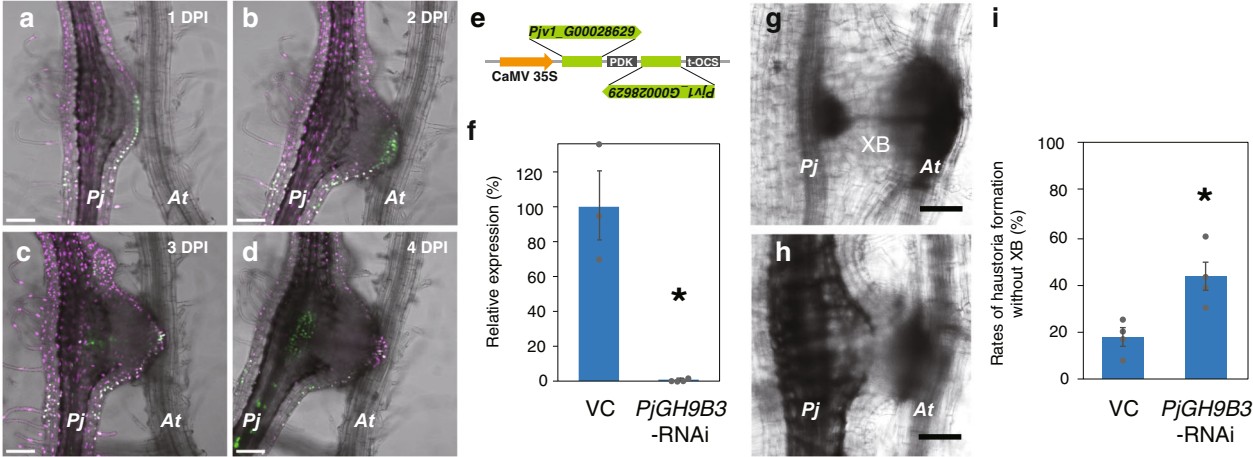

**Fig. 7 Involvement of *Glycosyl hydrolase 9B3* in establishment of parasitism of *P. japonicum*. a–d** Expression patterns of *PjGH9B3* promoter::*GFP* 1, 2, 3, and 4 days post infection (DPI). **e** RNAi targeting sequence for *PjGH9B3* and the construct. **f** Relative expression levels of *PjGH9B3* at 4 DPI (mean ± SE of three replicates). *PjGH9B3*-knockdown plants (*PjGH9B3*-RNAi) were categorized by the presence and absence of XB (XB and No XB, respectively). *PjUBC2* was used as a reference gene. Representative images of the haustoria that did (**g**) and did not (**h**) form a xylem bridge (XB) at 4 DPI. **i** Ratio of the haustoria that did not form a XB (mean ± SE of four replicates, *n* = 4–26). Dots indicate each data point. An asterisk indicates statistical significance (Welch's *t*-test, *P* < 0.05). VC, vector control. *Pj* *P. japonicum*, *At* *Arabidopsis*, XB xylem bridge. Scale bars, 100 μm.

capability of *P. japonicum* (Fig. 2c–g and Supplementary Data 1). Incompatibility of tissue adhesion was also observed in some occasions. In the case using Fabaceae plants as a grafting partner, two out of three species failed in successful grafting (Supplementary Data 1). Since these three species are not genetically distant from each other compared with other successful partner species tested in this study, these results represented that grafting compatibility of *P. japonicum* does not depend on phylogenetic relationships but on each plant's character, such as cell proliferation and differentiation abilities, or their combinations. Another observation was that *L. philippensis*, a nonparasitic Orobanchaceae species, failed to establish interfamily grafting (Supplementary Data 1). These facts imply that parasitic plants may have acquired a mechanism to reconstruct the cell walls at the interface with a broad range of plant species in the evolution of parasitism.

Involvement of cell wall modification in parasitism has been reported in various parasitic plant lineages[13,14,19,20]. This study identified a part of responsive genes for cell wall modification and performed functional analysis. Plant cells form primary and secondary cell walls. Primary cell walls are synthesized in cells of growing tissues and are composed predominantly of cellulose, pectic polysaccharides, and hemicelluloses such as xyloglucans. Secondary cell walls are formed in mature cells and are composed of cellulose, xylans, and lignin[28]. *P. japonicum* expresses β-1,4-glucanases of the GH9B3 family, which have secretory signal peptides and target glucan chains comprising cellulose[29,33,34] (Supplementary Fig. 4), at the periphery of the haustorium where parasites have direct physical contact with host tissue at 1–2 DPI (Fig. 7 and Supplementary Fig. 5). The *GH9B* gene family is highly conserved in plants[29]. Previous studies of microbial GH9 proteins have shown that these enzymes generally cleave the β-1,4-linkages of the glycosyl backbones of amorphous cellulose[33,34]. *CELLULASE 3* (*CEL3*) and *CEL5* are GH9B3 clade homologs in *Arabidopsis* that show cellulase activities and are considered to be involved in cell loosening and expansion in lateral roots and detachment of root cap cells[35–37]. We recently showed that *GH9B3* clade genes play roles in cell–cell adhesion in graft junctions of *Nicotiana*, and that of *Arabidopsis*[21]. Furthermore, this study showed that *P. japonicum* also achieved tissue adhesion during grafting with several distant plant species and expressed β-1,4-glucanases of the GH9B3 family (Figs. 2, 5, and 6). In parasitism, the knockdown of *PjGH9B3* genes

caused a defect in xylem bridge formation but not in the haustorium formation or morphology (Fig. 7f–i). Given that xylem bridge formation in haustorium starts after the parasitic intrusive cells reach the host vascular tissue[1], it is possible that *GH9B3*-silenced haustoria cannot reach the host vasculature likely due to the lack of sufficient enzymatic activity to loosen the host cell wall (Supplementary Fig. 5). Thus, *P. japonicum* activates conserved cell wall-degrading enzymes that target cellulose, the predominant cell wall polysaccharide of various cell types in plants. This phenomenon may partially account for the compatibility in tissue adhesion of parasitic plants with diverse host plant species. In addition, continuing expression of *PjGH9B3* could reflect its potential roles in the later steps of parasitism (Fig. 6b).

Gene duplication is involved in the evolution of parasitism. A previous study showed that duplication events occurred in more than half of the parasitic genes of Orobanchaceae species, which comprise a large number of genes annotated with GO terms associated with cell wall-modifying enzymes and peptidase activity[13]. Duplicated genes are potential drivers of functional innovation and adaptive evolution[38,39]. Notably, the number of *GH9B3* and *GH9B8* genes increases in Orobanchaceae parasites (*P. japonicum*, *S. asiatica,* and *S. hermonthica*) but not in non-parasites (*E guttata* and *L. philippensis*), suggesting that these genes may have been co-opted to plant parasitism in the Orobanchaceae (Fig. 6a and Supplementary Fig. 2). In contrast, no notable expansion was observed for other clades of the GH9 family (Fig. 6a and Supplementary Fig. 2). Gene duplication observed in the KAI2 strigolactone receptor family in *S. hermonthica* is also considered to be a driver for successful parasitism[4]. The number of *GH9B3* clade genes in *P. japonicum* is not notably higher compared with the KAI2 duplication event in *S. hermonthica*. In the case of *GH9B3* genes, four out of five homologous genes in *P. japonicum* were predicted to encode functional proteins in our alignment sequence analysis (Supplementary Fig. 4), but their expression patterns were temporally varied (Figs. 6b, 7a–d, and Supplementary Fig. 3a). Taken together, we assume that duplication of *GH9B3* and the resulting diversification in spatio-temporal expression are both important factors in the evolution of parasitism.

We narrowed the genes associated with tissue adhesion among distant species (Figs. 4 and 5) and identified a role in parasitism for one of candidate genes using *P. japonicum* system (Figs. 6 and 7).

Additional important components are likely to facilitate cell wall digestion and accomplish tissue adhesion. The dataset accumulated in the present study provides a foundation to identify such components involved in parasitic infection and/or grafting. *P. japonicum* is a useful model system because the seeds can germinate in the absence of a host plant, obtain transformants from hairy roots, parasitize the host, and can be grafted to the host plant species[16,22,23,40] (Figs. 1 and 2). Improved knowledge of the mechanisms responsible for parasitism could be applicable to suppress yield losses in crop cultivation caused by parasitic plants. The present study may provide an option, which might be especially effective against hemiparasites, to decrease parasitization by parasitic plants after germination by inhibiting the activities of secreted endo-β-1,4-glucanases. Several mono- and polysaccharides are reported to be inhibitors of cellulase[41]. Hence, a knowledge-based defense approach would further enhance crop security.

## Methods

**Plant materials**. For the grafting experiments, *P. japonicum* and *Arabidopsis thaliana* ecotype Columbia (Col–0) seeds were directly surface-sown on soil. Plants were grown at 22 °C under continuous light. Infection assay was performed as described previously[23]. Briefly, *Arabidopsis* seeds were surface-sterilized with 70% ethanol for 10 min, washed three times with sterile water, incubated at 4 °C for 2 days, and sown on half-strength Murashige and Skoog (1/2 MS) medium (0.8% agar, 1% sucrose, pH 5.8). Seedlings were grown vertically at 22 °C under long-day (LD) conditions (16 h light/8 h dark). *P. japonicum* seeds were surface-sterilized with 10% (v/v) commercial bleach solution (Kao, Tokyo, Japan) for 5 min, rinsed three times with sterilized water, and sown on 1/2 MS agar medium (0.8% agar, 1% sucrose). The agar plates were incubated at 4 °C in the dark overnight, then incubated at 25 °C under LD. Plants for the infection assay and the transformation experiment were cultured vertically and horizontally, respectively.

**Grafting**. For grafting of *P. japonicum*/*Arabidopsis*, 1–2-month-old *P. japonicum* and 1-month-old *Arabidopsis* plants were used. Wedge grafting was performed on the epicotyls, stems, petioles, or peduncles. For stock preparation, stems (or other organs) were cut with a 2–3-cm slit on the top. For scion preparation, the stem (around 7–10 cm from the tip of the stem) was cut and trimmed into a V-shape. The scion was inserted into the slit of the stock and wrapped with parafilm to maintain close contact. A plastic bar was set along the stock and the scion to support the scion and the graft. The entire scion was covered with a plastic bag, which had been sprayed inside with water beforehand. Grafted plants were grown for 7 days in an incubator at 27 °C under continuous light (~30 µmol m$^{-2}$ s$^{-1}$). After this period, the plastic bags were partly opened by cutting the bags and making holes for acclimation. The next day, the plastic bags were removed and the grafted plants were transferred to a plant growth room at 22 °C under continuous light conditions (~80 µmol m$^{-2}$ s$^{-1}$). All other plant materials for stem grafting used in this study are listed in Supplementary Data 1.

**Microscopy**. All the chemicals for staining were purchased from Sigma-Aldrich Co., Tokyo, Japan unless otherwise stated. Preparation of resin sections and observation by a transmission electron microscope were performed as previously described[21]. In this study, we showed other sections of the same grafted plant sample prepared in previous study[21]. To capture images of infection tissues or hand-cut grafted regions, a stereomicroscope (SZ61, Olympus, Tokyo, Japan) equipped with an on-axis zoom microscope (Axio Zoom.V16, Zeiss, Göttingen, Germany). To observe xylem tissues, phloroglucinol staining (Wiesner reaction) was performed by applying 18 µL of 1% phloroglucinol in 70% ethanol followed by the addition of 100 µL of 5 N hydrogen chloride to the section samples. To determine apoplastic transport, the stems of *Arabidopsis* stocks were cut and the cut edge was soaked in 0.5% toluidine blue solution for 4 h to overnight. To determine symplasmic transport, cut leaves of the *Arabidopsis* host or stock were treated with 0.01% 5(6)-CF diacetate. The CF fluorescence images were acquired by a fluorescence imaging stereomicroscope or observing emissions in the 490–530 nm range with excitation at 488 nm with a confocal laser scanning microscopy (LSM780, Zeiss). To examine graft junctions and visualize tissue adhesion, grafting was performed between a *P. japonicum* scion and a transgenic *Arabidopsis* stock, *RPS5A::LTI6b-tdTomato*, that ubiquitously expresses a plasma membrane-localized fluorescent protein[42]. Hand-cut cross-sections of the grafted stem regions were stained with 0.001% calcofluor white, which stains cellulose in plant cell walls. The fluorescence of tdTomato or calcofluor white was detected using a confocal laser scanning microscope (LSM710, Zeiss). A 555 nm laser and collecting emission spectrum of 560–600 nm were used for tdTomato and a 405 nm excitation laser and collecting emission spectrum of 420–475 nm were used for calcofluor white. The GFP or mCherry fluorescence images were acquired as described previously[22].

**Transcriptome analysis**. RNA-Seq was conducted for sequential samples of haustorial infection sites in the roots and of grafted regions on the stems as described previously[21,24,43]. Haustorial attachment/infection sites on the host roots and ~10–15 mm of graft junction or stem tissue at a similar location were sampled at the represented time points. For parasitism and grafting samples between *P. japonicum* and *Arabidopsis*, the sequence reads were mapped on the genome assembly using TopHat version 2.1.0 with default parameters (http://ccb.jhu.edu/software/tophat/). Uniquely mapped reads are counted using HTSeq (https://htseq.readthedocs.io/en/master/), and high homology genes that show more than ten total reads of *P. japonicum* or *Arabidopsis* samples mapped on *Arabidopsis* or *P. japonicum* genome, respectively, were removed. These mapped reads were normalized using the Bioconductor package EdgeR ver. 3.4.2 (https://bioconductor.org/packages/release/bioc/html/edgeR.html) with the trimmed mean of *M*-values method where reads were filtered based on the following criteria; there was at least one count per million in at least half of the samples. For the sample of grafting between *L. philippensis*, *Nicotiana* and *Arabidopsis*, the sequence reads were mapped on the genome assembly using HISAT2 version 2.1.0 (http://daehwankimlab.github.io/hisat2/). Gene expression levels, expressed as fragments per kilobase of transcript per million fragments mapped, were estimated using Cufflinks version 2.1.1 (http://cole-trapnell-lab.github.io/cufflinks/). The reference sequence used for mapping and the annotation file used were as follows: *P. japonicum* PjScaffold_ver1; *N. benthamiana* draft genome sequence v1.0.1, https://btiscience.org/our-research/research-facilities/research-resources/nicotiana-benthamiana/; *A. thaliana* TAIR10 genome release, https://www.arabidopsis.org; and *L. philippensis* LiPhGnB1, http://ppgp.huck.psu.edu. GO enrichment analysis was performed with DAVID (https://david.ncifcrf.gov) using *Arabidopsis* gene IDs. Transcriptome data of parasitism and grafting were used for PCA to compare the differences between samples. The Python version 3.7.4 and its library modules including NumPy (1.17.2), Pandas (0.25.1), SciPy (1.3.1), Matplotlib (3.1.1), seaborn (0.9.0), and scikit-learn (0.21.3) were used for PCA and hierarchical cluster classification. For SOM clustering, the parasitism was classified into nine clusters and the grafting was classified into 12 clusters. After clusters with similar patterns were paired, three clusters with increased expression during cell adhesion of both the parasitism and the grafting, and the clusters located before and after those were presented.

**Plasmid construction**. We used Golden Gate modular cloning to construct a vector to examine the expression pattern of *PjGH9B3* during infection[44]. The *PjGH9B3* promoter region (1899 bp upstream of the ATG start codon) was cloned into the vector pAGM1311 as two fragments and then combined into the vector pICH41295 to generate the promoter module. This module was assembled into the vector pICH47751 containing 3xVenus-NLS and AtHSP18.2 terminator[22], then subsequently further assembled into pAGM1311 with *pPjACT::3xmCherry-NLS*[22]. For RNAi experiments, we used the pHG8-YFP vector[45]. Target sequences, from 1175 to 1474 in coding sequence, were PCR-amplified from *P. japonicum* genomic DNA and cloned into the pENTR vector (Thermo Fisher Scientific, Waltham, MA, USA), then transferred into the pHG8-YFP vector by the Gateway reaction using LR Clonase II Plus enzyme (Thermo Fisher Scientific). All primers used in this paper are listed in Supplementary Data 11.

**P. japonicum transformation**. *P. japonicum* transformation was performed as previously described[36]. Six-day-old *P. japonicum* seedlings were sonicated and then vacuumed for 10 s and 5 min, respectively, in an aqueous suspension of *Agrobacterium rhizogenes* strain AR1193. After incubation in freshly prepared Gamborg's B5 medium (0.8% agar, 1% sucrose, 450 µM acetosyringone) at 22 °C in the dark for 2 days, seedlings were incubated in Gamborg's B5 medium (0.8% agar, 1% sucrose, 300 µg/ml cefotaxime) at 25 °C under LD.

**Parasitization assay and RT-qPCR**. Ten-day-old *P. japonicum* seedlings were transferred from 1/2 MS medium agar plates to 0.7% agar plates and incubated at 25 °C under LD for 2 days. Seven-day-old *Arabidopsis* seedlings were placed next to *P. japonicum* seedlings so that *P. japonicum* root tips could physically contact the *Arabidopsis* roots. For RNAi experiments, fresh elongated hairy roots were transferred to 0.7% agar plates and incubated at 25 °C under LD for 2 days. YFP- or mCherry-positive hairy roots were placed between thin agar layers (0.7%) and glass slides in glass-bottom dishes (Iwaki, Haibara, Japan), and incubated at 25 °C under LD overnight. Seven-day-old *Arabidopsis* seedlings were placed next to *P. japonicum* seedlings underneath agar layers so that *P. japonicum* root tips could physically contact the *Arabidopsis* roots. Xylem bridge formation and the expression pattern of *PjGH9B3* were examined using a confocal microscope (Leica SP5). After phenotyping xylem bridge formation, haustorial tissues were dissected and snap frozen in liquid nitrogen. Total RNA was extracted using the RNeasy Plant Mini kit (Qiagen, Hilden, Germany) and cDNA was synthesized using ReverTra Ace qPCR RT Master Mix (TOYOBO, Osaka, Japan). The RT-qPCR analyses were performed using a Stratagene mx3000p quantitative PCR system (with 50 cycles of 95 °C for 30 s, 55 °C for 60 s, and 72 °C for 60 s) with the Thunderbird SYBR qPCR Mix (TOYOBO). *PjUBC2* was used as an internal control as previously described[23]. The primer sequences used are shown in Supplementary Data 11. All experiments were performed with three independent biological replicates and three technical replicates.

**Statistics and reproducibility**. The details about statistics used in data analyses performed in this study are described in the respective sections of results and methods. For transcriptome analysis, we used four independent replicates of grafted or infected *P. japonicum* and *Arabidopsis*, including ten independent plants in each experiment. For *PjGH9B3*-knockdown experiments, the number of biological replicates and plant samples are given in figure legend.

**Reporting Summary**. Further information on research design is available in the Nature Research Reporting Summary linked to this article.

## Data availability

The RNA-Seq data are available from the DNA Data Bank of Japan (DDBJ; http://www.ddbj.nig.ac.jp/) under the accession number, DRA010010. All the source data underlying the graphs and charts are presented in Supplementary Data 2, 3, 12 and 13.

## Code availability

The details of publicly available program used in this study are described in the section of "Methods". No custom code or mathematical algorithm that is deemed central to the conclusions were used in this study.

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

## Acknowledgements

We thank S. Yoshida (Nara Institute of Science and Technology, Japan) for providing information relevant to the *P. japonicum* genome and Y. Tsuchiya (Nagoya University, Japan) for providing *L. philippensis* plants. We thank T. Shinagawa, H. Fukada, A. Ishiwata, and A. Shibata for technical assistance. This work was supported by grants from the Japan Society for the Promotion of Science Grants-in-Aid for Scientific Research (18KT0040, 18H03950, 19H05361 to M.N. and 15H05959, 17H06172 to K.S.),

the Canon Foundation (R17-0070 to M.N.), and the Japan Science and Technology Agency (START15657559 and PRESTO15665754 to M.N.).

## Author contributions

M.N., Y.I., K.S. and K.K. conceived the research and designed the experiments. M.N., K.O., and Y.S. performed the grafting experiments and morphological analysis. Y.I., T.S., S.C., and K.K. performed the RNA-Seq analysis. Y.I. performed the SOM clustering analysis. K.K. analyzed the transcriptome data. T.W. and S.O. performed functional analysis of the candidate genes. M.N. and K.S. supervised the experiments. K.K., T.W., K.S. and M.N. wrote the manuscript.

## Competing interests

The authors declare no competing interests.
