## [Peer Review file · Communications Biology]

Reviewers' comments:

Reviewer #1 (Remarks to the Author):

In this MS, Kurotani et al. found that the hemiparasite, *Phtheirospermum japonicum* (PJ), could be grafted with inter-family species, as either the rootstock or scion. PJ was naturally (parasitization) or artificially grafted (stem-stem grafting) onto *Arabidopsis*, and samples from different times were collected and used for in-depth comparative transcriptome analysis. Even though many genes were similarly regulated in both sets of samples, the transcriptome profiles of PJ sample from parasitization and grafting experiment were largely distinct. The authors also show that a GH9B gene is commonly upregulated in PJ-*Arabidopsis* parasitization and grafting transcriptomes, as well as in the transcriptomes from *Nicotiana*-*Arabidopsis* grafting junction. Finally, the expression pattern of PjGH9B3 was determined using a PjGH9Be promoter-Venus reporter gene and RNAi of PjGH9B3 indicated that this glycosyl hydrolase gene, possibly a β -1,4-glucanase, is a positive regulator of parasitism.

This work nicely demonstrates the involvement of β -1,4-glucanases in parasitism, which have been shown to be involved in tissue adhesion during artificial grafting, providing new insight into the mechanism underlying the establishment of parasitism.

I have only minor comments.

1. The RNA-seq samples were collected from the haustorial infestation and grafting regions. Thus, the RNA-seq data contain reads from both plant species, PJ and *Arabidopsis*. If I understood correctly, the reads were mapped to both genomes, and the *Arabidopsis*-specific reads and reads that could be mapped to both genomes were removed. The remaining reads were used for transcriptome analysis. If so, I would suggest adding a brief description of how *Arabidopsis* transcripts were filtered out in the Results, so that the readers can better understand the transcriptome analysis.
2. One to 7 days and one to 14 days samples respectively from PJ parasitizing on *Arabidopsis* and PJ-*Arabidopsis* grafting were collected for RNA-seq. Why were different time windows used for these samples? The reasons should be indicated.
3. It is interesting that PJ can even function as a rootstock or scion and can be grafted with plants from other families. Being a rootstock or scion, PJ has different success rates during grafting and PJ cannot be grafted with Fabaceae species. The reasons could be discussed.
4. In the Abstract, "symbiotic" could be misleading, could be replaced with "parasitic".
5. P8 L1-3. It is rather abrupt here that the transcriptomes from *Nicotiana*-*Arabidopsis* grafting system were used for further comparison. It would be better if the authors could elaborate the reasons.
6. P14 L9-11, it is important to know how the samples were collected. It would be better if the authors could indicate the details of sample collection, instead of citing other references.
7. P8 L9, the reference 20 is wrong.

Reviewer #2 (Remarks to the Author):

The manuscript by Kurotani et al. has investigated the amazing potential of the parasitic plant *Phtheirospermum japonicum* to facilitate interspecific cell-to-cell adhesion between its haustorial cells and cells of its host. The authors showed that this ability is not limited to cells of the haustorium but that also the shoot is graft-compatible with a wider range of graft partners than typically observed for other species. They further describe the identification of a secreted glycosyl hydrolase (and 8 other proteins) with overlapping expression in graft and infection sites and provide evidence that this glycosyl hydrolase is involved in the establishment of the xylem bridge between parasite and host, an event that marks a successful parasitic relationship.

The well-written manuscript altogether provides a lot of convincing data that is well presented and that represents a significant step forward in our understanding of how parasitic plants can

successfully attach to the vasculature of a broad host range.

I have only found a few minor issues that should be addressed before this manuscript is ready for publication.

1. Page 8, lines 3, 9, and 26: Please check that Reference 20 (Honaas et al. 2013) in these lines is correct. It seems to me that it is Ref. 21 (Notaguchi et al. 2020) that was meant here.
2. Page 9: Regarding the knock-down PjGH9B3-mutant, I wonder whether the expression of the other GH9B3 family members is somehow affected? It would be relevant to know whether the knockdown is specific for one gene only or affected the other related genes as well, or whether other GH9B3 family members may take over the role of the knocked-down gene, possibly with some delay, see also comment 4 below.
3. Page 11, line 9: "haustorium" (correct the typo)
4. Page 11, discussion of the haustorium in the knockdown PjGH9B3: I wonder if the authors also analyzed older haustoria of the genetically modified (knockdown) line? Could it be that the knockdown just influences the efficiency of an infection and that after a longer period of time the same outcome (i.e. xylem bridge formation) is observed?

Reviewer #3 (Remarks to the Author):

Brief summary of the manuscript

The manuscript by Kurotani et al. describes the results of grafting *Phtheirospermum japonicum*, a model parasitic plant of the family Orobanchaceae, on species of a variety of plant families, among them and *Arabidopsis*. They show that *P. japonicum* can successfully be grafted, both as rootstock and scion on different species while it can parasitize achieving successful tissue adhesion and vasculature connection. Excessive grafting experiments show that viable grafts of *P. japonicum* scions on *Arabidopsis* stems can be produced. They further used 9 more species as rootstocks in grafting tests while *P. japonicum* was also used as a rootstock. Successful grafting was observed. Then the authors compare the transcriptome of grafted *P.japonicum/Arabidopsis* grafting regions with *P. japonicum* parasitism sites at *Arabidopsis* roots and interfamily grafting between *Arabidopsis* and *Nicotiana*, and they show that 9 genes are commonly induced in these three datasets. Detailed transcriptomic analyses of graft and parasitism interfaces revealed three clusters of commonly distinct expression patterns that shared GO terms.

One of them, belongs to the family of Glycosyl hydrolase 9B family. GH9B3 from *Nicotiana* has been found before to be crucial for cell-cell adhesion in grafting. By expression analysis they show that this gene in *P. japonicum* follows the trend of other similar genes of the same plant family i.e. upregulation at the site of infection and after interfamily grafting. Knockdown of PjGH9B3 gene by RNAi experiments further suggest that this gene is possibly affecting the formation of xylem bridges between the host and the parasite, thus regulating infection process in a positive way.

Overall impression of the work

My overall impression is that the present work is novel, complete, and all the experiments are well executed. The results are presented clearly. The study contributes significantly to the amount of knowledge we have on grafting and parasitism. It also presents important common aspects and mechanisms by which these two processes are functioning. I believe that it contributes significantly to basic scientific understanding although I would like to see some more practical applications (possibly on plant grafting).

Specific comments, with recommendations for addressing each comment

1. Lines 11 – 14; this sentence needs re-writing.
2. Page 3; Lines 29 – Page 4; Lines 1; this sentence does not make sense and needs re-writing.
3. Page 8; Lines 5 – 7. "Belonging" needs to be substituted with "Belongs".
4. Page 10; Line 1. "Realize" isn't a suitable verb here, maybe authors want to replace it with "materialize" or "implement" (heterografting).

5. Page 11; Lines 28 – 30. Authors refer to the alignment of Supplementary Fig. 4 where five homologous genes of *P. japonicum* are aligned to 2 *Arabidopsis* genes. Yet in the text (line 29) they mention four out of five homologous genes in the Orobanchaceae parasitic plants. Maybe they need to specify that it is specifically the *P. japonicum* genes in the alignment?

Furthermore, in Figure 3b that depicts the expression patterns of genes after grafting and parasitism. Authors mention that genes PIN1, cyclin B1;2, PLL1, VND7 and OPS are all upregulated in a similar manner in both parasitism and grafting (genes that behaved similarly). Although I can see a similar pattern in the expression of OPS, VND7 and possibly PLL1, in PIN1 expression I see a reduction after 1 day of parasitism and then an increase (especially after the 2nd day) while in grafting there's an immediate increase in the expression after 1 day. Could the authors elaborate a bit more on why they interpret their results in this way?

To Reviewer #1

(Remarks to the Author):

In this MS, Kurotani et al. found that the hemiparasite, *Phtheirospermum japonicum* (PJ), could be grafted with inter-family species, as either the rootstock or scion. PJ was naturally (parasitization) or artificially grafted (stem-stem grafting) onto *Arabidopsis*, and samples from different times were collected and used for in-depth comparative transcriptome analysis. Even though many genes were similarly regulated in both sets of samples, the transcriptome profiles of PJ sample from parasitization and grafting experiment were largely distinct. The authors also show that a GH9B gene is commonly upregulated in PJ-*Arabidopsis* parasitization and grafting transcriptomes, as well as in the transcriptomes from *Nicotiana-Arabidopsis* grafting junction. Finally, the expression pattern of PjGH9B3 was determined using a PjGH9Be promoter-Venus reporter gene and RNAi of PjGH9B3 indicated that this glycosyl hydrolase gene, possibly a β -1,4-glucanase, is a positive regulator of parasitism.

This work nicely demonstrates the involvement of β -1,4-glucanases in parasitism, which have been shown to be involved in tissue adhesion during artificial grafting, providing new insight into the mechanism underlying the establishment of parasitism.

I have only minor comments.

We thank the reviewer for considering our studies providing new insight into parasitism.

1. The RNA-seq samples were collected from the haustorial infestation and grafting regions. Thus, the RNA-seq data contain reads from both plant species, PJ and *Arabidopsis*. If I understood correctly, the reads were mapped to both genomes, and the *Arabidopsis*-specific reads and reads that could be mapped to both genomes were removed. The remaining reads were used for transcriptome analysis. If so, I would suggest adding a briefly description of how *Arabidopsis* transcripts were filtered out in the Results, so that the readers can better understand the transcriptome analysis.

⇒ **Yes, the reviewer's understanding is correct. We described the filtering strategy of the sequence reads in Materials and Methods. We added a brief information also in Results (Page 6, Line 19–22).**

2. One to 7 days and one to 14 days samples respectively from PJ parasitizing on *Arabidopsis* and PJ-*Arabidopsis* grafting were collected for RNA-seq. Why were different time windows used for these samples? The reasons should be indicated.

⇒ **We appreciate your comment. We did not use the same timescale because observations of grafting between *N. benthamiana* and *Arabidopsis* and transcriptome analysis in previous study indicated that the time taken to establish grafting between distantly related plants was longer than the time taken to establish the parasitism. We added a text to explain this in Results (Page 6, Line 15–16).**

3. It is interesting that PJ can even function as a rootstock or scion and can be grafted with plants from other families. Being a rootstock or scion, PJ has different success rates during grafting and PJ cannot be grafted with Fabaceae species. The reasons could be discussed.

⇒ **Thank you for pointing out this curious observation. We are also interested in the reason why a part of Fabaceae species cannot be grafted with. Incompatibility can be occurred by several reasons for examples, less adhesion, less cell proliferation and less differentiation. At this moment, we don't have any evidence to distinguish these potential reasons and this could be a focus in the future research. We added a text in Discussion (Page 10, Line 22).**

4. In the Abstract, “symbiotic” could be misleading, could be replaced with “parasitic”.

⇒ **Thank you. We corrected the text (Abstract, Line 3).**

5. P8 L1-3. It is rather abrupt here that the transcriptomes from Nicotiana-Arabidopsis grafting system were used for further comparison. It would be better if the authors could elaborate the reasons.

⇒ **We added some texts about the reasons (Page 8, Line 8–10).**

6. P14 L9-11, it is important to know how the samples were collected. It would be better if the authors could indicate the details of sample collection, instead of citing other references.

⇒ **We added explanations (Page 14, Line 19–21).**

7. P8 L9, the reference 20 is wrong.

⇒ **Thank you. We corrected the mistake (Page 8, Line 10, 16 and Page 9, Line 3).**

To Reviewer #2

(Remarks to the Author):

The manuscript by Kurotani et al. has investigated the amazing potential of the parasitic plant *Phtheirospermum japonicum* to facilitate interspecific cell-to-cell adhesion between its haustorial cells and cells of its host. The authors showed that this ability is not limited to cells of the haustorium but that also the shoot is graft-compatible with a wider range of graft partners than typically observed for other species. They further describe the identification of a secreted glycosyl hydrolase (and 8 other proteins) with overlapping expression in graft and infection sites and provide evidence that this glycosyl hydrolase is involved in the establishment of the xylem bridge between parasite and host, an event that marks a successful parasitic relationship.

The well-written manuscript altogether provides a lot of convincing data that is well presented and that represents a significant step forward in our understanding of how parasitic plants can successfully attach to the vasculature of a broad host range.

I have only found a few minor issues that should be addressed before this manuscript is ready for publication.

We thank the reviewer for considering our studies represents a significant step forward in understanding of parasitism to a broad host range.

1. Page 8, lines 3, 9, and 26: Please check that Reference 20 (Honaas et al. 2013) in these lines is correct. It seems to me that it is Ref. 21 (Notaguchi et al. 2020) that was meant here.

⇒ **Thank you. We corrected the mistake (Page 8, Line 10, 16 and Page 9, Line 3).**

2. Page 9: Regarding the knock-down PjGH9B3-mutant, I wonder whether the expression of the other GH9B3 family members is somehow affected? It would be relevant to know whether the knockdown is specific for one gene only or affected the other related genes as well, or whether other

GH9B3 family members may take over the role of the knocked-down gene, possibly with some delay, see also comment 4 below.

⇒ **We appreciate reviewer’s careful consideration on our experiments. Because the primary DNA sequences of the *PjGH9B3* homologs are quite different from other family members, the possibility of cross-silencing is extremely low in principle. It is possible that positive or negative feedback regulation of other genes could occur as the reviewer mentioned. Here, however, the major question we tested was whether *PjGH9B3* contributes to establishment of parasitism and we believe that the question is properly addressed. The downstream event due to GH9B3 KO would be investigated in the future analysis when clear KO system is available for *P.japonicum* (such as CRSPR in a stable transformation).**

3. Page 11, line 9: “haustorium” (correct the typo)

⇒ **Thank you. We corrected the mistake (Page 11, Line 17).**

4. Page 11, discussion of the haustorium in the knockdown *PjGH9B3*: I wonder if the authors also analyzed older haustoria of the genetically modified (knockdown) line? Could it be that the knockdown just influences the efficiency of an infection and that after a longer period of time the same outcome (i.e. xylem bridge formation) is observed?

⇒ **Thank you for thoughtful comment. We have not seen any example in that earlier defects in haustorium development recovered later time points. In other words, once a developmental process in haustorium formation ceases then it does not proceed further. Of course, testing a role of *PjGH9B3* in the older stage in parasitism would be an interest in the future study but it requires specific promoters for conditional silencing. In this study, we preferred to focus on the function of *PjGH9B3* in establishment of the first adhesion. We added a sentence to mention its potential roles in the later steps of parasitism in Discussion (Page 11, Line 23–24).**

To Reviewer #3

(Remarks to the Author):

Brief summary of the manuscript

The manuscript by Kurotani et al. describes the results of grafting *Phtheirospermum japonicum*, a model parasitic plant of the family Orobanchaceae, on species of a variety of plant families, among them and *Arabidopsis*. They show that *P. japonicum* can successfully be grafted, both as rootstock and scion on different species while it can parasitize achieving successful tissue adhesion and vasculature connection. Excessive grafting experiments show that viable grafts of *P. japonicum* scions on *Arabidopsis* stems can be produced. They further used 9 more species as rootstocks in grafting tests while *P. japonicum* was also used as a rootstock. Successful grafting was observed. Then the authors compare the transcriptome of grafted *P. japonicum*/*Arabidopsis* grafting regions with *P. japonicum* parasitism sites at *Arabidopsis* roots and interfamily grafting between *Arabidopsis* and *Nicotiana*, and they show that 9 genes are commonly induced in these three datasets. Detailed transcriptomic analyses of graft and parasitism interfaces revealed three clusters of commonly distinct expression patterns that shared GO terms.

One of them, belongs to the family of Glycosyl hydrolase 9B family. GH9B3 from *Nicotiana* has been found before to be crucial for cell-cell adhesion in grafting. By expression analysis they show that this gene in *P. japonicum* follows the trend of other similar genes of the same plant family i.e. upregulation at the site of infection and after interfamily grafting. Knockdown of PjGH9B3 gene by RNAi experiments further suggest that this gene is possibly affecting the formation of xylem bridges between the host and the parasite, thus regulating infection process in a positive way.

Overall impression of the work

My overall impression is that the present work is novel, complete, and all the experiments are well executed. The results are presented clearly. The study contributes significantly to the amount of knowledge we have on grafting and parasitism. It also presents important common aspects and mechanisms by which these two processes are functioning. I believe that it contributes significantly to basic scientific understanding although I would like to see some more practical applications (possibly on plant grafting).

We thank the reviewer for considering our studies contribute significantly to the knowledge of grafting and parasitism. Since part of the mechanism of the establishment of grafting has been elucidated, together with the previously presented study, we believe that future improvements in practical grafting technology can be expected by utilizing these gene functions.

Specific comments, with recommendations for addressing each comment

1. Lines 11 – 14; this sentence needs re-writing.

⇒ **Thank you. We corrected the sentences (Page 3, Line 11–14).**

2. Page 3; Lines 29 – Page 4; Lines 1; this sentence does not make sense and needs re-writing.

⇒ **Thank you. We corrected the sentence (Page 3, Line 29 – Page 4, Line 1).**

3. Page 8; Lines 5 – 7. “Belonging” needs to be substituted with “Belongs”.

⇒ **Thank you. We corrected the mistake (Page 8, Line 13).**

4. Page 10; Line 1. “Realize” isn’t a suitable verb here, maybe authors want to replace it with “materialize” or “implement” (heterografting).

⇒ **Thank you. We corrected the mistake (Page 10, Line 9).**

5. Page 11; Lines 28 – 30. Authors refer to the alignment of Supplementary Fig. 4 where five homologous genes of *P. japonicum* are aligned to 2 *Arabidopsis* genes. Yet in the text (line 29) they mention four out of five homologous genes in the Orobanchaceae parasitic plants. Maybe they need to specify that it is specifically the *P. japonicum* genes in the alignment?

⇒ **Yes. We completely agree with the reviewer and we corrected the manuscript accordingly (Page 12, line 8).**

Furthermore, in Figure 3b that depicts the expression patterns of genes after grafting and parasitism. Authors mention that genes PIN1, cyclin B1;2, PLL1, VND7 and OPS are all upregulated in a similar manner in both parasitism and grafting (genes that behaved similarly). Although I can see a similar pattern in the expression of OPS, VND7 and possibly PLL1, in PIN1 expression I see a reduction after 1 day of parasitism and then an increase (especially after the 2nd day) while in grafting there’s an immediate increase in the expression after 1 day. Could the authors elaborate a bit more on why they interpret their results in this way?

⇒ **We agree with the reviewer’s view point. Figure 3 shows the overall pattern of gene expression in parasitism and grafting and it shows that they are not very similar. In**

Figure 3B, which shows the patterns of individual genes previously reported, we did not mention the details, but pointed out similarities by making a broad determination of whether the trend was an "upward trend" or "no variation" or a "downward trend". We expect that the difference in expression patterns could be reasoned by the difference of phenomena in different organs. We added a short interpretation (Page 7, Line 4–6).

REVIEWERS' COMMENTS:

Reviewer #1 (Remarks to the Author):

The authors have addressed all my concerns and comments.

Reviewer #2 (Remarks to the Author):

In their revised version of the manuscript, the reviewers have responded to all issues that I had. There is only one more typo that I stumbled upon: On page 11, line 17, it should either read "the haustorium" (with a definite article) or haustoria (plural). Apart from that I can only congratulate the authors on their excellent piece of work.

Reviewer #3 (Remarks to the Author):

I have nothing else to add to my previous review when it comes to the Brief Summary of the Manuscript and the Overall Impression of the Work. I appreciate the authors effort to correct/address/amend every single comment and I'm satisfied by the authors replies to my specific comments.

Reviewer #1 (Remarks to the Author):

The authors have addressed all my concerns and comments.

Reviewer #2 (Remarks to the Author):

In their revised version of the manuscript, the reviewers have responded to all issues that I had. There is only one more typo that I stumbled upon: On page 11, line 17, it should either read "the haustorium" (with a definite article) or haustoria (plural). Apart from that I can only congratulate the authors on their excellent piece of work.

Thank you. We corrected the mistake (Page 11, Line 16).

Reviewer #3 (Remarks to the Author):

I have nothing else to add to my previous review when it comes to the Brief Summary of the Manuscript and the Overall Impression of the Work. I appreciate the authors effort to correct/address/amend every single comment and I'm satisfied by the authors replies to my specific comments.